# Decrease in All-Cause 30-Day Mortality after Bacteraemia over a 15-Year Period: A Population-Based Cohort Study in Denmark in 2000–2014

**DOI:** 10.3390/ijerph18115982

**Published:** 2021-06-02

**Authors:** Mona Katrine Alberthe Holm, Filip Jansåker, Kim Oren Gradel, Rikke Thoft Nielsen, Christian Østergaard Andersen, Jens Otto Jarløv, Henrik Carl Schønheyder, Jenny Dahl Knudsen

**Affiliations:** 1Department of Clinical Microbiology, Copenhagen University Hospital Hvidovre, Kettegaard Alle 30, 2650 Hvidovre, Denmark; RTN@ssi.dk (R.T.N.); Christian.Oestergaard.Andersen@regionh.dk (C.Ø.A.); 2Department of Clinical Microbiology, Copenhagen University Hospital, Rigshospitalet, 2100 Copenhagen N, Denmark; filip_jansaker@hotmail.com (F.J.); Inge.Jenny.Dahl.Knudsen@regionh.dk (J.D.K.); 3Center for Primary Health Care Research, Department of Clinical Sciences Malmö, Lund University, 214 28 Lund, Sweden; 4Center for Clinical Epidemiology, Odense University Hospital, 5000 Odense, Denmark; kim.gradel@rsyd.dk; 5Research Unit of Clinical Epidemiology, Department of Clinical Research, University of Southern Denmark, 5230 Odense, Denmark; 6Department of Infectious Disease Epidemiology and Prevention, Statens Serum Institut, 2300 Copenhagen S, Denmark; 7Department of Clinical Microbiology, Copenhagen University Hospital, Herlev and Gentofte, 2730 Herlev, Denmark; Jens.Otto.Jarloev@regionh.dk; 8Department of Clinical Microbiology, Aalborg University Hospital, 9000 Aalborg, Denmark; hcs@rn.dk; 9Department of Medicine, Aalborg University, 9220 Aalborg, Denmark

**Keywords:** bacteraemia, bloodstream infection, population-based study, mortality, epidemiology

## Abstract

Introduction: Bacteraemia is a frequent infectious condition that strongly affects morbidity and mortality. The incidence is increasing worldwide. This study explores all-cause 30-day mortality after bacteraemia in two out of Denmark’s five healthcare regions with approximately 2.4 million inhabitants. Methods: Clinically significant bacteraemia episodes (*n* = 55,257) were identified from a geographically well-defined background population between 2000 and 2014, drawing on population-based data regarding bacterial species and vital status. All-cause 30-day mortality was assessed in relation to bacteraemia episodes, number of patients with analysed blood cultures and the background population. Results: We observed a decreasing trend of all-cause 30-day mortality between 2000 and 2014, both in relation to the number of bacteraemia episodes and the background population. Mortality decreased from 22.7% of the bacteraemia episodes in 2000 to 17.4% in 2014 (annual IRR [95% CI]: 0.983 [0.979–0.987]). In relation to the background population, there were 41 deaths per 100,000 inhabitants in 2000, decreasing to 39 in 2014 (annual IRR [95% CI]: 0.988 [0.982–0.993]). Numbers of inhabitants, bacteraemia episodes, and analysed persons having BCs increased during the period. Conclusions: All-cause 30-day mortality in patients with bacteraemia decreased significantly over a 15-year period.

## 1. Introduction

Bacteraemia, or bloodstream infection, is a frequent finding often associated with substantial mortality and contributes to a large healthcare burden worldwide [1]. Bacteraemia is confirmed by the growth of bacterial microorganisms from a blood culture (BC) obtained from a patient with clinical signs of infection after ruling out contamination. The most common causative agents are *Escherichia coli* followed by *Staphylococcus aureus* [2]. Especially adults over the age of 65 years and infants below the age of one are at higher risk of dying within the first 30 days after bacteraemia [3].

In the last three decades, an increase in the bacteraemia burden has been reported [1,3,4,5,6]. To assess the implication of this increase, it is important to study the evolution of mortality after bacteraemia over time, to monitor and evaluate the effect of current treatment strategies. Over time, this will allow continuous treatment optimisation.

Determining the exact cause of death is difficult, which may lead to cause of death in bacteraemia patients to be recorded as, e.g., urinary tract infection, pneumonia, or endocarditis, resulting in an underestimation of bacteraemia as the cause of death [7,8]. Population-based studies are the reference standard method determining the incidence of infectious diseases within a geographical area [9]. In Denmark, all occurrences of positive BCs can be identified from well-defined geographic regions and the date of death can be retrieved from national databases [10]. This enables reliable data on both the incidence of bacteraemia and all-cause 30-day mortality after bacteraemia [8].

In the last few decades, an increase in antimicrobial resistance (AMR) has also been registered and higher rates of mortality have been found when infected with resistant bacterial strains. Data from the European Antimicrobial Resistance Surveillance Network showed substantial geographic differences in the proportion of resistance to various classes of antibiotics, with lower levels in northern Europe and higher levels in southern Europe [11]. The current low levels of AMR in Denmark are not expected to contribute substantially to our results; thus, it is crucial to understand the future threat of the progressing AMR and irrational use of antibiotics in Europe for optimal future prevention strategies [12].

The aim of this study was to determine the all-cause 30-day mortality after first-time bacteraemia in two of Denmark’s five healthcare regions from 2000 through 2014, in relation to number of bacteraemia episodes, analysed number of patients having BCs, and background population.

## 2. Materials and Methods

### 2.1. Setting

In Denmark, all residents are provided free healthcare and treatment through tax-supported public hospitals. All Danish residents are registered with a personal 10-digit civil registration number (CPR) that incorporates the date of birth and sex and enables linkage between all administrative registries [10,13]. 

### 2.2. Bacteraemia Study Cohort

This study was a population-based cohort study identifying first-time bacteraemia patients within two of Denmark’s five regions: the Capital Region and the North Denmark Region, with approximately 2.4 million (out of Denmark’s approximately 5.7 million) inhabitants. Patients were identified through the three departments of clinical microbiology (DMCs) serving this population: Aalborg University Hospital, Herlev Hospital, and Hvidovre Hospital [14]. 

Data of included BCs were collected from 1 January 2000 through 31 December 2014. First-time positive BC findings were obtained from the DACOBAN database, including vital status (alive, dead, or emigration) [13]. The total number of patients with analysed BCs, both positive and negative, were retrieved directly from the laboratory information systems at the three DMCs. All data were anonymised before starting analysis and CPR numbers were encrypted. 

Bacteraemia was defined according to globally defined criteria, i.e., if pathogenic bacteria were detected once or bacteria generally considered contaminants (*Cutibacterium acnes*, *Micrococcus* spp. coagulase-negative staphylococci, *Corynebacterium* spp., *Bacillus* spp.) were detected twice within a five-day period from the first BC collection [15,16]. 

### 2.3. Identification and Susceptibility Testing

BCs were taken upon clinical suspicion of infection with a minimum of 30 mL blood in total and incubated, using either BACTEC^TM^ (BD, Franklin Lakes, NJ, USA) or BacT/ALERT^TM^ (Biomérieux, France) [17]. Pathogens were identified using conventional methods and susceptibility tested according to the EUCAST methodology [18]. The Departments of Clinical Microbiology used the available equipment for mass spectrometry (Malditof) after 2009 and directly on positive BCs in the last years. 

### 2.4. Empiric Treatment

The primary empiric treatment used during the period for suspected severe bacterial infection with unknown focus was piperacillin/tazobactam 4 g/0.5 g or cefuroxime 1.5 g every 8 h +/− gentamicin 5 mg/kg once daily, or ampicillin 2 g every 6 h and gentamicin 5 mg/kg once daily, dose adjustable for weight and kidney function according to guideline [19].

### 2.5. Background Populations and the Statistical Analysis

Initially, we computed the geographic background population for each year within the two regions by the help of Statistics Denmark [20]. From 2000–2006, Denmark consisted of 13 counties which were merged to five larger regions on 1 January 2007 [21]. Another change occurred in 2012 when a department of clinical microbiology in Northern Zealand (Hillerød) was closed, and bacteraemia data from patients in the Northern Zealand area were transferred to the department of clinical microbiology in Herlev from September 2003 and onwards. Consequently, 2003 data are excluded from the Capital Region in Figure 1. Due to facilitation issues of BC handling in the North Denmark Region, BCs from three municipalities (Thisted, Sydthy, and Morsø) were sent to another department of clinical microbiology (Viborg) outside the North Denmark Region for analysis and the background population in the North Denmark Region has been adjusted accordingly. Furthermore, BCs analysed at the highly specialised referral Hospital, Rigshospitalet, were excluded.

We computed the incidence rates (IRs) of bacteraemia episodes, and all-cause 30-day mortality in relation to the background population. We further computed the IRs of bacteraemia episodes in relation to analysed BCs as well as 30-day mortality in relation to both analysed BCs and bacteraemia episodes. 

Initially, we studied graphs of these IRs for changes occurring from 2000 through 2014. To corroborate these assessments, we performed Poisson regression analysis if the goodness-of-fit test’s *p*-value was >0.05 [22]. If not, we either split into smaller time periods or used a negative binomial regression model, which allows overdispersion. 

Regardless of model, we report results as incidence rate ratios (IRRs) with 95% confidence intervals (CIs); if CIs did not overlap 1, the results were considered statistically significant. 

We further assessed whether there was an increasing or decreasing trend of median age (using a non-parametric trend test [23]) or percentage of females (using the Chi-square test for trend), with *p* < 0.05 considered significant. 

Microsoft Excel software was used for the descriptive analysis and Stata vs. 14 software (StataCorp., College Station, TX, USA) for the regressions.

## 3. Results

### 3.1. Study Population

We identified 55,257 first-time bacteraemia episodes from 1 January 2000 through 31 December 2014, 29,244 (53%) episodes occurred in males and 26,013 (47%) in females (Appendix A). A significant decline was found in female patients throughout the period, from an approximately even gender distribution in 2000/2001 to approximately 45% females in 2013/2014 (*p* < 0.0001, data not shown). From age 60 and onwards, the female background population increasingly exceeds the male population (Appendix A). The median age was 72 years, with no clear trends of decline or increase during the 15-year period and with little variation between years (ranging from 72.2 to 73.8 years). 

### 3.2. Calculations in Relation to Background Population

The background population size increased by 39%, from 1,674,697 in 2000 to 2,330,462 in 2014 (Figure 1A). 

In the Capital Region, 1493 patients had analysed BCs per 100,000 inhabitants in the year 2000, increasing to 2412 patients in 2014. In the North Denmark Region, 1695 patients had BCs analysed per 100,000 inhabitants in the year 2000, increasing to 2337 in 2014, amounting to a total increase of 55% in analysed BCs per 100,000 inhabitants within the two regions (Figure 1B).

Confirmed first-time bacteraemia episodes per 100,000 inhabitants were 172 in the Capital Region in 2000, increasing to 230 in 2014, while numbers were 206 episodes in the North Denmark Region in 2000 with a slight decrease to 200 in 2014. A combined increase of 22.9% was registered within the two regions (Figure 1C).

All-cause 30-day mortality per 100,000 inhabitants showed a slightly decreasing tendency throughout the 15-year period, from 41 per 100,000 in 2000 to 39 in 2014, which is a 6% combined decrease within the two regions (annual IRR [95% CI]: 0.988 [0.982–0.993]) (Figure 1D).

### 3.3. Total Number of Patients and Mortality

In 2000, a total of 25,875 patients had analysed BCs increasing to 55,775 (116%) in 2014 (Figure 2A). First-time bacteraemia was confirmed in 3031 patients in 2000, increasing to 5182 (71%) in 2014, of which 688 patients were recorded dead at 30 days in 2000, increasing to 904 (31%) patients in 2014 (Figure 2C). All-cause 30-day mortality in relation to first-time bacteraemia decreased by 30%, from 22.7% in 2000 to 17.4% in 2014 (Figure 2D) (annual IRR [95% CI]: 0.983 [0.979–0.987], *p* < 0.0001). 

## 4. Discussion

Our findings confirmed that the number of patients with analysed BCs more than doubled between 2000–2014 (Figure 2A) and the number of first-time bacteraemia patients increased by 71% (Figure 2B). The number of patients who died within 30 days after confirmed bacteraemia increased by 31% (Figure 2C), while the all-cause 30-day mortality decreased from 23% in 2000 to 17% (Figure 2D) in 2014 within two out of Denmark’s five regions. Initially, we hypothesised that the decrease in all-cause 30-day mortality rate could potentially be the result of accelerated blood culturing. To elucidate this further, we computed the all-cause 30-day mortality per 100,000 inhabitants, and thus, encountered a significant 6% decrease over the 15 years. Despite the 6% decline in deaths per 100,000 inhabitants, the increase in bacteraemia burden is clear. Out of 100,000 inhabitants, an overall 23% increase in bacteraemia patients was registered during the 15-year period. It has been theorised that the increase in bacteraemia patients could be a result of an increasing aging population [24]; however, in our study, and in the study by Søgaard et al. [1] as well as Nielsen et al. [25], the median age remained stable around 72 years between 1992 and 2014 (data not shown). The increase could also be explained by increased sepsis awareness [26] as well as continuous improvements in BC methodology resulting in improved bacterial detection [27]. 

In the past three decades, increasing bacteraemia incidence has been reported in studies from other developed countries in Europe, Australasia, and North America [1,3,4,5,6,7,9,28], several of which are population-based studies comparable to ours [7,9,28]. In concordance with our experience, and partially covering the current data from the North Denmark Region, Søgaard et al. [1] registered a doubling number of persons with analysed BCs, a significant increase in bacteraemia patients and total number of deaths, as well as a 30-day all-cause mortality decrease from 22.7% to 20.6% between 1992 and 2006. The all-cause 30-day mortality decrease may be a result of increased awareness, construction of clearer guidelines, improvements in urinary catheter quality programs, optimisation of empirical antibiotic treatment, introduction of pneumococcal conjugate vaccine, as well as administration of annual influenza vaccines [26,29,30,31,32,33,34]. 

In a systematic review of population-based studies from multiple developed countries from 1970–2013, Laupland reported the bacteraemia incidence ranging between 80 and 189 per 100,000 inhabitants, with increasing tendency each year [9], where we found an overall increase from 181 patients per 100,000 inhabitants in the year 2000 to 222 patients in 2014 (Figure 1C). Kern and Goto et al. [7,28] found a similar evolution over time. Nielsen et al. [25], however, concluded contradicting results. Nielsen et al. found an overall 23.3% decrease in first-time bacteraemia patients in their population-based study between 2000–2008 in the Danish county Funen. Nielsen et al. concluded that their conflicting results are caused by multiple factors, such as the difficulty of direct comparisons between different studies, due to a lack of standardised references in BC methodology, sampling rates, and bacteraemia definition, especially for contaminants. We speculate that the fall described by Nielsen et al. might be due to the shorter range of years included in their study compared with other population-based studies. When observing our incidence (Figure 1C), we also see a temporary fall between 2004–2008, before continuing the increasing tendency. 

Among bacteraemia patients, we found a linearly increasing male dominance from an approximate even sex distribution in 2000 to 55% in 2014 (Appendix A). Likewise, male dominance (52.5%) was previously described by Steckelberg et al. [35] in an American population based study of 650 bacteraemia patients between 2003–2005, and by Nielsen et al. [25] (54%) among 7786 bacteraemia patients from the Danish county Funen, as well as by Søgaard et al. [1] (53.8–54.7%). The male bacteraemia patient dominance is interesting when considering that the female sex distribution is increasing from age 60 and onwards in the background population. According to Statistics Denmark [20] the background population constituted of 54% females in 2008 and 53% in 2014 (Appendix A).

Our study suggests that we have become better at treating bacteraemia, but there is still room for improvements to bring down mortality further. We encourage continuous optimisation of empiric antimicrobial therapy, improved infection hygiene, optimised tools for earlier diagnosis, and susceptibility testing of pathogens for quicker targeted treatment. Early diagnostics and targeted treatment of, e.g., urinary tract infection, can prevent the bacteria from spreading from the bladder to the blood while allowing optimal treatment with as narrow spectrum antibiotic as the bacteria allow. In respiratory infections, quick testing distinguishes viral from bacterial infection and can eliminate unnecessary antibiotic consumption if virus is detected. 

As bacteraemia is treated with antibiotics, awareness of the AMR evolution is crucial as AMR inevitably will become an increasing problem in the future. Machowska et al. clarified factors concerning the rapid evolution of AMR and the threat irrational use of antibiotics poses. Lack of public knowledge and awareness, access to antibiotics without prescription, leftover antibiotics, knowledge, attitude, and perception of prescribers and dispensers, inadequate medical training as well as lack of rapid and sufficient diagnostic tests are areas of major concern [12]. A general public questionnaire from 2016, based on 27,969 respondents from 28 European member states, with the aim to determine the Europeans antibiotic knowledge, found that less than half of the Europeans were aware that antibiotics are ineffective against viruses [36]. According to Eurobarometer, only 32% of Europeans stated receiving correct information about antibiotics from a doctor, while the rest received information from TV advertisements, news, internet, pharmacists, friends, and family. A concerning increase in the use of antibiotics without a prescription, purchased online, or in another country has been reported [36]. An Italian national cross-sectional survey among 1179 young doctors proved lack of AMR knowledge [37], while in London, only 5–13% of junior doctors thought their previous education on the use of antibiotics was sufficient [38]. This supports the importance of better training and improved medical education on AMR as well as the implementation of antimicrobial stewardship programs and increased raised awareness among the European general population.

To optimise public health further, governments should ensure proactive and effective antibiotic control and restrict consumption by only allowing antibiotics upon prescription, as well as prioritising research and development of new antibiotics before we run out of treatment options [12].

### Strengths and Limitations

The strengths of this study include the population-based design, reliable data, complete follow-up, high number of bacteraemia episodes, and a large cohort of approximately 2.4 million inhabitants during a 15-year period. Limiting factors include the continuous change in the background population, which complicated the determination of the exact number of inhabitants as well as the possibility the patients dying from underlying conditions other than bacteraemia.

## 5. Conclusions

In conclusion, this study demonstrated a considerable increase in the overall bacteraemia burden during the 15-year study period in Denmark, while all-cause 30-day mortality after first-time bacteraemia decreased. Even though the all-cause 30-day mortality has decreased, bacteraemia is still a resource-demanding and potentially deadly condition. To reduce the burden of bacteraemia in the future, we suggest focusing on quick diagnostics, targeted antibiotics, improved hygiene, and antibiotic education among healthcare providers and increased AMR awareness in the general public. 

## Figures and Tables

**Figure 1 ijerph-18-05982-f001:**
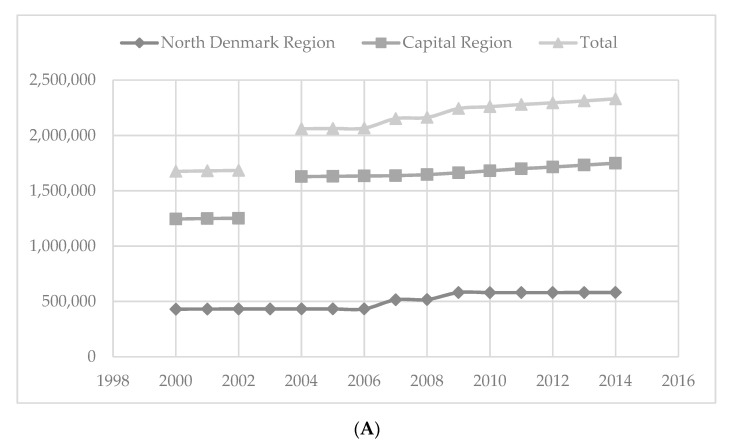
(**A**) Background population. (**B**) Analysed BCs per 100,000 inhabitants. (**C**) First-time bacteraemia episodes per 100,000 inhabitants. (**D**) Deaths at 30 days per 100,000 inhabitants.

**Figure 2 ijerph-18-05982-f002:**
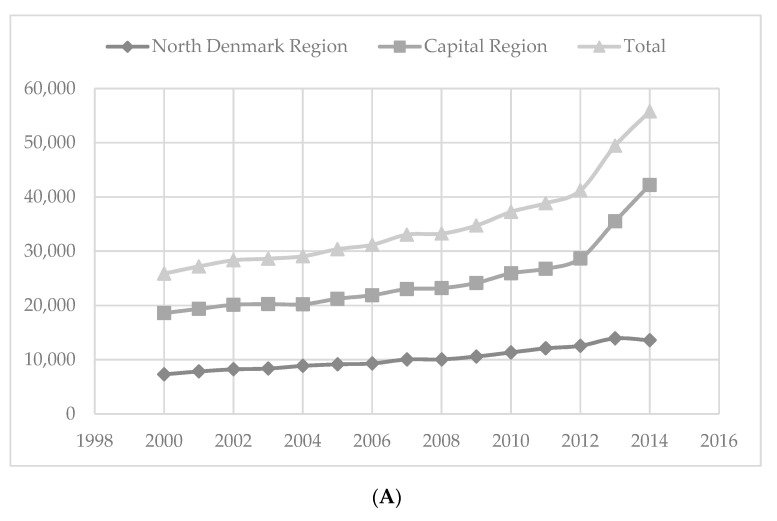
(**A**) Number of BCs analysed. (**B**) First-time bacteraemia episodes. (**C**) Death episodes at 30 days. (**D**) All-cause 30-day mortality in percentage.

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
