# Peer review of "Decrease in All-Cause 30-Day Mortality after Bacteraemia over a 15-Year Period: A Population-Based Cohort Study in Denmark in 2000–2014"

_ijerph, 2021, doi:10.3390/ijerph18115982_

Round 1

Reviewer 1 Report

This study by Holm et al. explores all-cause 30-day mortality after bacteraemia in two out of Denmark’s five healthcare regions with approximately 2.4 million inhabitants. The work is meaningful. But I don’t think this is a cohort study, as the authors stated in the title. I think of this study more as a trend report analysis of routine work.

Author Response

Dear reviewer #1

Thank you for reading our manuscript and for taking your time to review it.

We have carefully considered your comments regarding cohort study or not. We believe the study is a cohort study since the characteristic of a cohort study is that subjects are identified at one point in time when they do not have the outcome of interest, and the incidence of the outcome is compared among exposed and unexposed. In our study we followed all Danish residents within the two regions since year 2000 and monitored who got first-time bacteraemia over a 15-year period. Thereby we consider this study to be classified as a cohort study.

Furthermore, some points were marked, which can or must be improved, but without indulging specifically into these. Accordingly, we have tried our best and we hope this has improved the manuscript.

The sections have now been amended, clarified and enhanced in terms of why the study and monitorization of bacteraemia is important as well as something about antimicrobial resistance.

Reviewer 2 Report

I read with great interest the paper. I find it well wrote and with good idea research. Below my suggestions

  1. Introduction: add data on antimicrobial resistance in Europe and in your country
  2. Methods and results are well wrote
  3. Discuss : discuss also the role of Procalcitonin as predictive factors of outcome and new drugs against gram negative bacteria. Furthermore, add the role of education especially of young medical doctors to correct antibiotic prescription (see and cite if you wantItalian young doctors' knowledge, attitudes and practices on antibiotic use and resistance: A national cross-sectional survey. J Glob Antimicrob Resist. 2020 Dec;23:167-173.). Discuss also the role of infection prevention control
  4. In conclusion give some public health proposal that came from your interesting paper

Author Response

Dear reviewer #2

Thank you for reading our manuscript and for taking your time to review it.

1: Thank you for this point. We have added data on antimicrobial resistance in Europe and Denmark.
2: Thank you.
3:

  • Procalcitonin as a predictive factor for sepsis outcome is an important and interesting subject. Unfortunately, we do not have any data on procalcitonin on our bacteraemia patients and believe procalcitonin is outside of the scope of this paper.
  • We have included the importance of continuous development of new antibiotics in the bottom of discussion, which we agree is of importance to emphasize.
  • The importance of education has now been included.
  • Infection prevention control has also been included.

4: Thank you again, this has now been provided.

Round 2

Reviewer 1 Report

The manuscript has been greatly improved.

Author Response

Thank you once again for reviewing our manuscript. We have carefully fine-tuned spelling and grammar and believe it has been optimised satisfactory now.